# Longitudinal association between mental health and future antibiotic prescriptions in healthy adults: Results from the LOHAS

Kentaro Tochitani [1,2], Shungo Yamamoto[1,2], Tsukasa Kamitani[1,3], Hajime Yamazaki[4], Shunichi Fukuhara[4,5], Yosuke Yamamoto [1] *

1 Department of Healthcare Epidemiology, School of Public Health in the Graduate School of Medicine, Kyoto University, Kyoto, Japan, 2 Department of Infectious Diseases, Kyoto City Hospital, Kyoto, Japan, 3 Institute for Health Outcomes and Process Evaluation Research (iHope International), Kyoto, Japan, 4 Section of Clinical Epidemiology, Department of Community Medicine, Graduate School of Medicine, Kyoto University, Kyoto, Japan, 5 Center for Innovative Research for Communities and Clinical Excellence (CIRC[2]LE), Fukushima Medical University, Fukushima, Japan

* yamamoto.yosuke.5n@kyoto-u.ac.jp

**Data Availability Statement:** The underlying data was obtained from the collaboration with the local government and contains sensitive information on

## Abstract

### Objectives

To investigate the association of mental health and subjective physical functioning with future antibiotic prescriptions.

### Design

Prospective cohort study.

### Setting

A rural town in Japan.

### Participants

Participants who completed the baseline survey (2008–2010) of the Locomotive Syndrome and Health Outcomes in the Aizu Cohort Study (LOHAS) were recruited. Participants were limited to those without comorbidities according to the Charlson comorbidity index. Participants using antibiotics at baseline were excluded. Mental health and physical functioning were assessed using the Mental Health and Physical Functioning domains of the Short-Form 12 Health Survey, and depressive symptoms were assessed using the Mental Health Inventories at baseline.

### Main outcome measures

The main outcome was antibiotic prescriptions found in claims data during 1 year after the baseline survey.

individuals including gender, age, and self-reported social data, and sharing these data openly is prohibited by the contract with the local government. Data requests may be sent to iHope international (research@i-hope.jp). The authors did not have any special privileges to access the data and others would be able to access these data in the same manner as the authors.

**Funding:** The authors received no specific funding for this work.

**Competing interests:** The authors have declared that no competing interests exist.

## Results

A total of 967 participants were included in the analysis, and 151 (15.6%) participants with at least one missing variable for the confounding factors were excluded, leaving 816 participants for the primary analysis. Among the 816 participants, 65 (8.0%) were newly prescribed at least one antibiotic during the 1-year follow-up period. The most frequently prescribed antibiotics were third-generation cephalosporins (44 prescriptions; 35.5%), macrolides (28 prescriptions; 22.6%), and quinolones (23 prescriptions; 18.6%). A multivariable logistic regression analysis showed an association between higher mental health scores and future antibiotic prescriptions (adjusted odds ratio [AOR], 1.40 per 1 standard deviation [SD] increase; 95% confidence interval [CI], 1.03–1.90), whereas no significant relationship was observed between Physical Functioning scores and future antibiotic prescriptions (AOR, 0.95 per 1 SD increase; 95% CI, 0.75–1.22). During the secondary analysis, adults with depressive symptoms were less likely to be prescribed antibiotics (AOR, 0.27; 95% CI, 0.11–0.70).

## Conclusions

Better mental health was associated with increased future antibiotic prescriptions for healthy community-dwelling Japanese adults, suggesting that mentally healthier adults could be a target population for reducing antimicrobial use.

## Introduction

Antimicrobial resistance (AMR) has become a global problem in recent years. Currently, the number of deaths attributed to AMR is approximately 70,000 per year worldwide, and it is predicted to increase to 10 million by 2050 unless measures are taken [1]. To prevent AMR, it is necessary to use antibiotics appropriately and reduce their use. The World Health Organization launched a global action plan to combat antimicrobial-resistant bacteria in 2015, and it asked member countries to approve national action plans within 2 years [2]. In 2016, the government of Japan launched a national action plan, and the goal was to reduce antimicrobial use by 33% by the year 2020 [3].

Reducing antibiotic prescriptions involves understanding associated factors, mainly medical staff factors and patient factors. Medical staff factors include age or sex of the physicians, practice volume, and regional characteristics [4, 5]. Similarly, patient factors may include age, sex, smoking status, socioeconomic status, and comorbidities [6–8]. Regarding factors associated with mental health, psychological stress might be associated with an increased occurrence of infectious diseases [9, 10], but there have been no studies of whether people with lower quality of life (QOL) are likely to be prescribed antibiotics.

This study aimed to analyze the longitudinal associations of mental health and subjective physical functioning of healthy community-dwelling Japanese adults with their future antibiotic prescriptions to clarify the individual characteristics associated with the increase in antibiotic prescriptions.

## Methods

### Design and setting

This was a secondary analysis of a prospective cohort study that involved participants in the Locomotive Syndrome and Health Outcomes in the Aizu Cohort Study (LOHAS). The

LOHAS is an ongoing population-based cohort study evaluating the association between physical dysfunction and clinical outcomes such as cardiovascular disease, QOL, medical costs, and mortality. In the LOHAS, health-related QOL was assessed at baseline using self-administered questionnaires between 2008 and 2010, and data were linked to annual health examinations conducted in the local municipalities. The LOHAS participants comprised residents of two municipalities (Tadami and Minamiaizu) in Fukushima Prefecture, Japan, who were older than 40 years of age and receiving regular health examinations conducted by the local government annually. The planned follow-up period was 10 years, and all data were linked to administrative data offered by the municipalities, including medical records and death certificates, to evaluate clinically relevant outcomes. The design of the LOHAS is described in detail elsewhere [11]. The present study enrolled participants from the LOHAS baseline survey.

All participants provided written informed consent, and the study was approved by the institutional review boards of the Fukushima Medical University School of Medicine and the Kyoto University School of Medicine.

## Study population

This study included only adults who lived in Tadami because administrative claims data were available only for Tadami from 2008 to 2010 during the study period. Participants were also limited to those without comorbidities according to the Charlson comorbidity index (CCI) scores.

Participants were enrolled in the survey during the first visit year from 2008 to 2010. We excluded participants who received antibiotics at baseline. A prescription at baseline was defined as the presence of any antibiotic prescription 2 months prior to completing the questionnaire. We also excluded participants who died or moved during the observation period.

## Data preparation

We retrospectively analyzed data from the administrative health insurance claims database of Tadami. The database consisted of medical and pharmacy claims provided by government insurers. Data regarding patient demographics (year and month of birth and sex), diagnosis, date of diagnosis, medical procedures, and medications were available monthly. Claims from each medical facility were registered at the end of the month. The diagnosis was recorded by the physicians of each health facility and coded according to the International Statistical Classification of Diseases and Related Health Problems, 10th Revision (ICD-10).

We extracted the date (year and month), type, amount, and duration of antibiotic prescriptions from the insurance claims database. In the database, the name of the infectious disease newly recorded during the antibiotic prescription month was assumed as the name of the disease for which the antibiotics were prescribed because the diagnosis leading to the prescription of antibiotics was not explicitly presented in the database.

In accordance with previous studies, we also extracted from the database the names of diseases based on the CCI using ICD-10 codes [12, 13]. Active diseases up to 1 year prior to completion of the questionnaire were included in the baseline CCI.

## Exposures

The main exposures were subject-perceived mental health and physical functioning at baseline assessed using the mental health and physical functioning domains of the Japanese version of the Short-Form (SF) 12 Health Survey (SF-12) (version 2) [14, 15]. Both mental health and physical functioning scores were standardized based on Japanese representative samples, with higher values indicating better QOL [15].

During the secondary analysis, we used the presence or absence of depressive symptoms at baseline as exposures. Depressive symptoms were assessed using the five-question Mental Health Inventories (MHI-5), which has been validated and widely used to screen participants suspected of having depressive symptoms. The MHI-5 score was originally derived from the mental health score of the SF-36, which is comparable with other SF tools. In this study, in accordance with a previous study, we converted the mental health score of the SF-12 to that of the SF-36 and dichotomized it using a cutoff value of ≤60 to define moderate or severe depressive symptoms [16].

## Outcomes

We defined the follow-up period as 1 year from the date of answering the questionnaire. The primary outcome was receiving an antibiotic prescription during the observation period. Two Japanese board-certified infectious disease physicians reviewed the claims database and excluded prophylactic antibiotics for surgery and trauma and antibiotics for *Helicobacter pylori* eradication. Surgery, trauma, and *H. pylori* infection were identified from the medical procedures and disease names.

We also described each antibiotic class according to the Anatomical Therapeutic Chemical (ATC) classification system [17]. We assumed that oral third-generation cephalosporins accounted for most of the cephalosporins used in Japan [18]; hence, we divided cephalosporins into three groups, first-generation and second-generation cephalosporins, oral third-generation cephalosporins, and other cephalosporins. We also described infectious disease names corresponding to prescribed antibiotics in accordance with previously reported classifications [19, 20].

## Measurements of potential confounding variables

Data regarding sociodemographic variables (age, sex, working status, and whether the participants lived alone or with families) and health-related behaviors (smoking status and alcohol consumption) were obtained using a self-reported questionnaire. Smoking was classified into three categories: current, former, or never. Alcohol consumption was classified into two categories: every day or sometimes and rarely or never. CCI scores were collected from the claims database as mentioned.

## Statistical analysis

Baseline characteristics are presented using standard descriptive statistics: medians (interquartile ranges) for continuous variables and percentages for categorical variables. During the primary analysis, we examined the association of continuous SF-12 mental health and physical functioning scores with antibiotic prescriptions during the 1-year follow-up period. Odds ratios (ORs) and 95% confidence intervals (CIs) for the risk of antibiotic prescriptions were estimated using the multivariable logistic regression model. In the logistic regression model, we adjusted for clinically relevant confounding factors such as age, sex, living alone, working status, smoking status, and alcohol consumption. Missing values of the SF-12 were imputed following the method of the scoring manual [15]. During the secondary analysis, a logistic regression analysis was performed to investigate the association between depressive symptoms at baseline and antibiotic prescriptions, with adjustment for the same confounders used during the primary analysis. Regarding participants who died or relocated during the follow-up period, those who showed the occurrence of the outcome before death or relocation were excluded from the analysis.

All analyses were conducted using Stata version 14.2 (StataCorp LP, College Station, TX). P<0.05 (two-tailed) was considered statistically significant.

### Sensitivity analysis

We performed a sensitivity analysis to determine the robustness of the results. Missing values of covariates were handled with multiple imputations, and all variables included in the primary analysis were used in the imputations model to generate 20 datasets. The results of each imputation dataset were integrated based on Rubin's rules [21].

## Results

From 2008 to 2010, a total of 984 residents without comorbidities who lived in Tadami were enrolled in this study. Among them, 17 (1.7%) met the exclusion criteria (received antibiotics at baseline, 7; died, 8; relocated, 2). Finally, 967 participants were included in the analysis (Fig 1).

During the primary analysis, 151 (15.6%) participants with at least one missing variable for confounding factors were further excluded, leaving 816 participants. The average age was 62.3 years (standard deviation [SD], 12.1 years), and 39.2% were male. The mean mental health and physical functioning scores were 52.4 (SD, 9.4) and 49.1 (SD, 11.2), respectively. The baseline characteristics of the participants are shown in Table 1. A total of 130 antibiotic prescriptions were prescribed for 65 participants (8.0%), with at least one antibiotic prescribed for each participant during the 1-year follow-up period. During the primary analysis, the most frequently prescribed antibiotics were third-generation cephalosporins (44; 35.5%), macrolides (28; 22.6%), and quinolones (23; 18.6%) (Table 2). The most common diagnoses were urinary tract infections (32; 25.8%), pneumonia (16; 12.9%), miscellaneous bacterial infections (15; 12.1%), viral upper respiratory infections (14; 11.3%), and pharyngitis (12; 9.7%) (Table 3).

During the multivariable logistic regression analysis with adjustment for confounding factors, higher mental health scores were associated with a greater likelihood of future antibiotic prescriptions (adjusted OR, 1.40 per 1 SD increase; 95% CI, 1.03–1.90; p = 0.03) (Table 4). In contrast, no significant association was observed between physical functioning scores and future antibiotic prescriptions (adjusted OR, 0.95 per 1 SD increase; 95% CI, 0.75–1.22; p = 0.71).

During the secondary analysis, depressive symptoms were associated with fewer future antibiotic prescriptions (adjusted OR, 0.27; 95% CI, 0.11–0.70; p = 0.007) (Table 5).

### Sensitivity analysis

The sensitivity analysis using the multiple imputation approach showed that the adjusted ORs of future antibiotic prescriptions per 1 SD increase in mental health and physical functioning scores were 1.26 (95% CI, 0.96–1.66; p = 0.09) and 0.94 (95% CI, 0.76–1.16; p = 0.55), respectively (S1 Table).

Multiple imputation performed during the secondary analysis also revealed that the adjusted OR of future antibiotic prescriptions for adults with depressive symptoms compared to those without was 0.44 (95% CI, 0.21–0.90; p = 0.03) (S2 Table).

## Discussion

This longitudinal study showed that among Japanese adults without comorbidities, those with poor mental health were less likely to receive future antibiotic prescriptions. Furthermore,

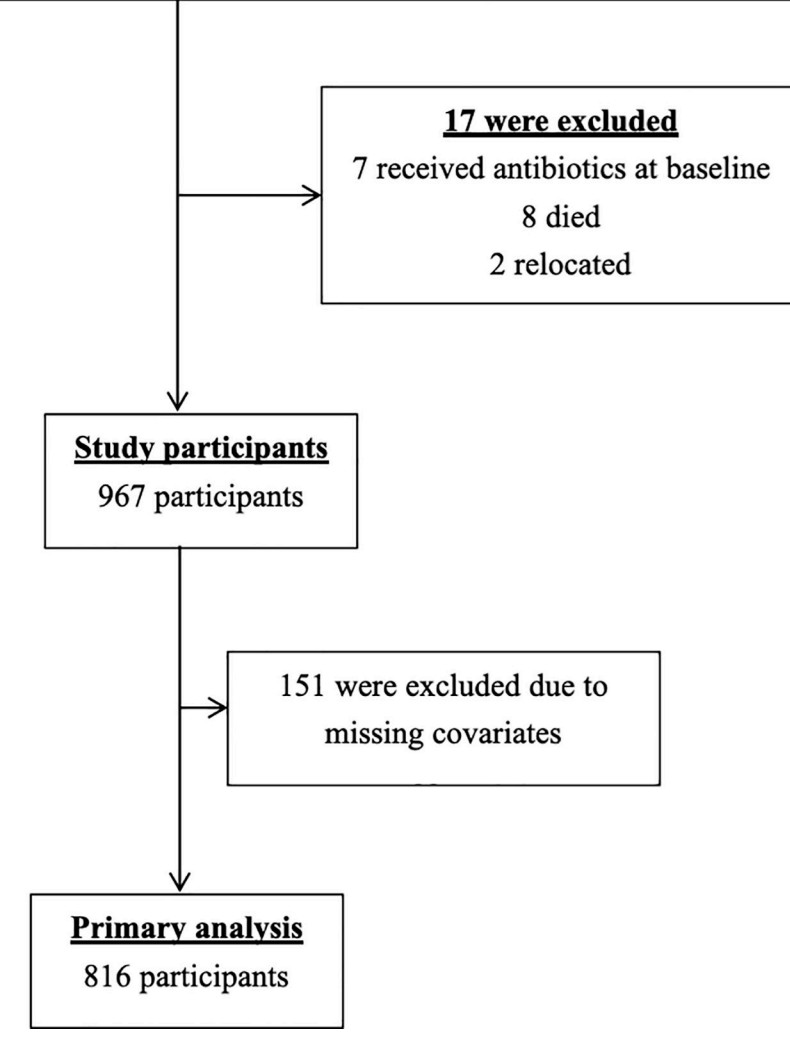

**Fig 1. Flow chart of the study participants.**

subjective physical functioning was not significantly associated with future antibiotic prescriptions.

To our knowledge, this is the first study to show a longitudinal association between subjective health states and future antibiotic prescriptions. Some observational studies have shown an association between psychological stress and an increased occurrence of respiratory tract infections [10, 22, 23]. An association between depressive disorders and poor clinical outcomes of pneumonia patients has also been reported [24]. In a recent large cohort study, mental illness was reported to be associated with the incidence of life-threatening infectious diseases [25]. Furthermore, one study revealed the association between posttraumatic stress disorder and the incidence of miscellaneous infectious diseases [26]. Therefore, depressive symptoms might be associated with increased antibiotic prescriptions. However, the results of the present study did not support such a hypothesis.

**Table 1. Characteristics of study participants.**

| | Study participants n = 967 | | Participants in the primary analysis n = 816 |
| --- | --- | --- | --- |
| | | Missing | |
| Age, mean (SD) | 63.4 (12.1) | 4 | 62.3 (12.1) |
| Male, n (%) | 377 (39.1) | 4 | 320 (39.2) |
| Occupation, n (%) | | 10 | |
| Yes | 548 (57.3) | | 477 (58.5) |
| No | 409 (42.7) | | 339 (41.5) |
| Living alone, n (%) | | 7 | |
| Yes | 84 (8.7) | | 71 (8.7) |
| No | 876 (91.3) | | 745 (91.3) |
| Smoking, n (%) | | 28 | |
| Current smoker | 144 (15.4) | | 133 (16.3) |
| Former smoker | 207 (22.0) | | 179 (21.9) |
| Never smoker | 588 (62.6) | | 504 (61.8) |
| Alcohol consumption, n (%) | | 137 | |
| Every day or sometimes | 387 (46.6) | | 378 (46.3) |
| Rarely or never | 443 (53.4) | | 438 (53.7) |
| SF-12 MH score, mean (SD) | 52.3 (9.5) | 7 | 52.4 (9.4) |
| SF-12 PF score, mean (SD) | 48.4 (11.9) | 4 | 49.1 (11.2) |
| Antibiotic prescription, n (%) | | 0 | |
| Yes | 74 (7.7) | | 65 (8.0) |
| No | 893 (92.3) | | 751 (92.0) |

SD, standard deviation; SF-12 MH, Short-Form 12 Health Survey Mental Health domain; SF-12 PF, Short-Form 12 Health Survey Physical Functioning domain

The unexpected association between depressive symptoms and decreased antibiotic prescriptions in this study might have been due to the excessive use of antibiotics among mentally healthy adults. A Japanese claims database-based study of antibiotics used by patients with acute nonbacterial upper respiratory tract disease showed that men in their 20s and 30s were most often prescribed antibiotics [27]. The study suggested that healthier people tend to visit the hospital more frequently for the treatment of infectious diseases and tend to receive unnecessary antibiotic prescriptions, which supports the speculation that inappropriate antibiotics might have been prescribed to mentally healthy residents without comorbidities in the present

**Table 2. Prescribed antibiotics according to antibiotic class.**

| Antibiotic class | Study participants (n = 967) 141 prescriptions, n (%) | Participants in the primary analysis (n = 816) 124 prescriptions, n (%) |
| --- | --- | --- |
| Penicillin | 7 (5.0) | 6 (4.8) |
| First-generation and second-generation cephalosporins | 9 (6.4) | 9 (7.3) |
| Third-generation cephalosporins | 56 (39.7) | 44 (35.5) |
| Other cephalosporins | 3 (2.1) | 3 (2.4) |
| Carbapenems | 5 (3.6) | 5 (4.0) |
| Macrolides | 31 (22.0) | 28 (22.6) |
| Quinolones | 23 (16.3) | 23 (18.6) |
| Tetracyclines | 3 (2.1) | 2 (1.6) |
| Aminoglycosides | 4 (2.8) | 4 (3.2) |

**Table 3. Antibiotic prescriptions for infectious disease diagnoses.**

| Diagnoses | Study participants (n = 967) 141 prescriptions, n (%) | Participants in the primary analysis (n = 816) 124 prescriptions, n (%) |
|---|---|---|
| Miscellaneous bacterial infections | 16 (11.4) | 15 (12.1) |
| Pneumonia | 17 (12.1) | 16 (12.9) |
| Abdominal infections | 6 (4.3) | 5 (4.0) |
| Orthopedic infections | 4 (2.8) | 4 (3.2) |
| Urinary tract infections | 32 (22.7) | 32 (25.8) |
| Pelvic inflammatory diseases | 1 (0.7) | 1 (0.8) |
| Gastrointestinal infections | 3 (2.1) | 2 (1.6) |
| Skin, cutaneous, and mucosal infections | 11 (7.8) | 11 (8.9) |
| Suppurative otitis media | 1 (0.7) | 1 (0.8) |
| Pharyngitis | 16 (11.4) | 12 (9.7) |
| Viral upper respiratory infections | 22 (15.6) | 14 (11.3) |
| Bronchitis | 7 (5.0) | 6 (4.9) |
| Unknown | 5 (3.5) | 5 (4.0) |

study. In contrast, people with poor mental health may be less likely to receive antibiotic prescriptions because their health conditions can be a hindrance to visiting the hospital or they inadequately treated their infections because of their mental problems. In the present study, subjective physical functioning was not associated with future antibiotic prescriptions, which could have been because we limited our sample to healthy participants without comorbidities, most of whom had no decline in physical functioning. The relationship between subjective physical functioning and future antibiotic prescriptions might vary among people with multiple comorbidities. However, analyzing such a relationship was difficult because we only used data from an insurance claims database. Future studies are required to examine associations

**Table 4. Primary analysis adjusted odds ratios for antibiotic prescriptions.**

| | Participants in the primary analysis (n = 816) | |
|---|---|---|
| | Adjusted OR [95% CI] | p value |
| SF-12 MH score (per 1 SD) | 1.40 [1.03–1.90] | 0.03 |
| SF-12 PF score (per 1 SD) | 0.95 [0.75–1.22] | 0.71 |
| Age (per year) | 1.01 [0.98–1.03] | 0.58 |
| Sex, female (vs. male) | 0.95 [0.52–1.74] | 0.86 |
| Occupation | | |
| Yes (vs. no) | 1.06 [0.60–1.86] | 0.84 |
| Living alone | | |
| Yes (vs. no) | 1.95 [0.93–4.11] | 0.08 |
| Smoking status | | |
| Never and former smoker | Ref | |
| Current smoker | 0.71 [0.31–1.62] | 0.41 |
| Alcohol consumption | | |
| Rarely or never | Ref | |
| Every day or sometimes | 1.23 [0.70–2.15] | 0.47 |

OR, odds ratio; CI, confidence interval; SD, standard deviation; SF-12 MH, Short-Form 12 Health Survey Mental Health domain; SF-12 PF, Short-Form 12 Health Survey Physical Functioning domain; Ref, reference

**Table 5. Secondary analysis adjusted odds ratios for antibiotic prescriptions.**

| | Participants in the primary analysis (n = 816) | |
| --- | --- | --- |
| | Adjusted OR [95% CI] | p value |
| Depressive symptoms* with (vs. without) | 0.27 [0.11–0.70] | 0.007 |
| SF-12 PF score (per 1 SD) | 0.96 [0.75–1.22] | 0.73 |
| Age (per year) | 1.01 [0.99–1.03] | 0.45 |
| Sex, female (vs. male) | 0.91 [0.50–1.68] | 0.77 |
| Occupation | | |
| Yes (vs. no) | 1.09 [0.62–1.92] | 0.76 |
| Living alone | | |
| Yes (vs. no) | 1.92 [0.91–4.05] | 0.09 |
| Smoking status | | |
| Never and former smoker | Ref | |
| Current smoker | 0.73 [0.32–1.68] | 0.46 |
| Alcohol consumption | | |
| Rarely or never | Ref | |
| Every day or sometimes | 1.21 [0.69–2.11] | 0.51 |

*Depressive symptoms indicated by an SF-12 MH score ≤60.

OR, odds ratio; CI, confidence interval; SD, standard deviation; SF-12 PF, Short-Form 12 Health Survey Physical Functioning domain; Ref, reference

between subjective health conditions and antibiotic prescriptions using data from chart reviews.

Regarding antibiotic use, this study showed that the most frequently prescribed antibiotics were third-generation cephalosporins, macrolides, and quinolones. This finding is consistent with those of previous Japanese studies [18, 20, 27, 28].

The strengths of this study include the use of a large sample of community-dwelling older adults, including both men and women, and the use of a longitudinal study design. We also used large and accurate antibiotic prescription datasets from an administrative claims database.

This study also had some limitations. First, although this was a population-based study, it was based on samples from only one town in Japan. Further studies are needed to ascertain the external validity of the findings. Second, data regarding the comorbidities could not be properly collected because the claims data might have included inaccurate names of diseases. Third, although a multiple imputation method was used for missing data based on the assumption of "missing at random," the exact reasons underlying the missing data are unknown. However, the results of the sensitivity analysis were similar to those of the primary analysis, thus supporting the robustness of the main results. Fourth, a selection bias might have occurred because some residents refused to participate in this study. Participation in this study was voluntary; therefore, it could be assumed that a higher number of residents with poorer mental health might have refused. Unfortunately, we could not know the exact number of residents who refused to participate. Fifth, measurement biases might have occurred because we could not verify the exact reasons for prescribing antibiotics or if the prescription was actually followed. However, we believe such a bias could occur regardless of mental health problems and we are assuming that it has little impact on our research. Finally, as a general limitation of observational studies, it was impossible to adjust for unknown confounding factors.

In conclusion, this study showed an association between better mental health and an increase in antibiotic prescriptions for community-dwelling Japanese adults without

comorbidities. This finding implies that mentally healthy residents without comorbidities could become a target population for reducing antimicrobial use.

## Supporting information

**S1 Table. Adjusted odds ratios for antibiotic prescriptions, primary analysis, and sensitivity analysis.**
(DOCX)

**S2 Table. Adjusted odds ratios for antibiotic prescriptions, secondary analysis, and sensitivity analysis.**
(DOCX)

## Acknowledgments

We would like to thank the staff of the public offices of Tadami and Minami-Aizu for their assistance with locating the participants and scheduling examinations and the participants of the LOHAS for their cooperation. We would like to thank Editage (www.editage.com) for English language editing.

## Author Contributions

**Conceptualization:** Kentaro Tochitani, Yosuke Yamamoto.

**Data curation:** Kentaro Tochitani, Yosuke Yamamoto.

**Formal analysis:** Kentaro Tochitani, Yosuke Yamamoto.

**Investigation:** Kentaro Tochitani.

**Methodology:** Kentaro Tochitani, Yosuke Yamamoto.

**Project administration:** Tsukasa Kamitani, Shunichi Fukuhara, Yosuke Yamamoto.

**Resources:** Kentaro Tochitani, Tsukasa Kamitani, Yosuke Yamamoto.

**Software:** Kentaro Tochitani.

**Supervision:** Shungo Yamamoto, Tsukasa Kamitani, Hajime Yamazaki, Shunichi Fukuhara, Yosuke Yamamoto.

**Validation:** Kentaro Tochitani, Yosuke Yamamoto.

**Visualization:** Kentaro Tochitani.

**Writing – original draft:** Kentaro Tochitani.

**Writing – review & editing:** Kentaro Tochitani, Shungo Yamamoto, Tsukasa Kamitani, Hajime Yamazaki, Shunichi Fukuhara, Yosuke Yamamoto.

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
