## [Decision Letter · Decision Letter 0]

26 Jun 2020

PONE-D-20-10789

Longitudinal association between mental health and future antibiotic prescriptions in healthy adults: Results from the LOHAS

PLOS ONE

Dear Dr. Yamamoto,

Thank you for submitting your manuscript to PLOS ONE. After careful consideration, we feel that it has merit but does not fully meet PLOS ONE’s publication criteria as it currently stands. Therefore, we invite you to submit a revised version of the manuscript that addresses the points raised during the review process.

The manuscript by Yamamoto et al. is well assessed by the two reviewers. However, there are still minor revisions in the present form. Read carefully the comments and respond to them appropriately.

We look forward to receiving your revised manuscript.

Kind regards,

Masaki Mogi

Academic Editor

PLOS ONE

Journal Requirements:

Reviewers' comments:

Reviewer's Responses to Questions

**Comments to the Author**

1. Is the manuscript technically sound, and do the data support the conclusions?

Reviewer #1: Yes

Reviewer #2: Yes

2. Has the statistical analysis been performed appropriately and rigorously? 

Reviewer #1: Yes

Reviewer #2: Yes

3. Have the authors made all data underlying the findings in their manuscript fully available?

Reviewer #1: No

Reviewer #2: Yes

4. Is the manuscript presented in an intelligible fashion and written in standard English?

Reviewer #1: Yes

Reviewer #2: No

5. Review Comments to the Author

Reviewer #1: Thank you for giving me the opportunity to review this manuscript, which describes the association between mental health and prescription of antibiotics.

The authors found that individuals with poorer mental health were less likely to be prescribed antibiotics, compared to individuals with better mental health. I believe the manuscript is well written, I enjoyed reading it, and all the analysis and explanations are clearly described. I have a few comments that might help the authors improve this manuscript.

P5 L95: The authors call this design “retrospective” but data were collected prospectively, so I suggest to reword in order to not confuse the reader.

P5 L 113: It would be useful to describe properly the selection of the study population. How many people were invited to participate? How many declined? How many excluded because of antibiotics at baseline? Or because of comorbidity? A flowchart would be very useful.

P6 L122: The authors excluded participants who died or moved without any prescription. I believe this is methodologically wrong. In my opinion, the best option would be to include everyone and censor them at time of dead/emigration. Then the analysis could be performed with a Poisson model or a similar model that allows for different follow-up time in different individuals. Alternatively (if death or emigration is rare), one might exclude all those who emigrate or die (although this might bias the results). However, the authors used one approach for those with antibiotics, and another for those without. In any case, I think the same approach should be used for all study participants (this also refers to P9 L193).

P6 L133: The authors used the prescription database to identify antibiotics, but it is not clear where the disease (assumed to be treated by the antibiotics) was obtained from. Is this from another database? It would be interesting to see the concordance between the two data sources: how many are in both, how much time apart the two occurrences are, etc.

P9 L203: It would be useful to add a reference to Rubin’s rules for those unfamiliar with multiple imputation.

P15 L323: The results of this study are clear: those with poorer mental health are less likely to be prescribed antibiotics. Have the authors considered looking only at infections? Is it possible that the rate of infections is the same but not treatment not adequate for those with mental health problems?

P16 L338: “The absence of any significant association” should be rephrased. There is a significant association, in the other direction.

P17 L363: I think the discussion would benefit of some limitations related to the internal validity of the study. For example, is there any potential selection bias (the flow diagram mentioned above would help assess that)? Could there be some misclassification of mental health status or antibiotic use? What happens with the difference between antibiotics prescription and actually taken?

Finally, I have some general comments. Why did the authors decide to exclude individuals with comorbidities? Would not be possible to investigate those? Or adjust for comorbidities in the analyses? Also, what is the main idea behind excluding patients with antibiotics at baseline? I would understand this approach if causal conclusions had to be taken, for example which exposures to change to prevent future antibiotic use. However, in this study, the main aim is to describe whether individuals with poorer/better mental health are less/more likely to use antibiotics, could this question be answered with a cross-sectional design, in which establishing a temporal relation is not the main aim?

Reviewer #2: Please carefully revise the manuscript for English grammar. Otherwise I very much enjoyed reading this study, it is the first study to examine the association between mental health and antibiotic use in the community setting, a topic I have very often wondered about, thank you for your work, it should be a welcome addition to the literature.

6. PLOS authors have the option to publish the peer review history of their article (what does this mean?). If published, this will include your full peer review and any attached files.

Reviewer #1: **Yes: **Oleguer Plana-Ripoll

Reviewer #2: **Yes: **Tony Velkov

---

## [Author Response · Author response to Decision Letter 0]

9 Aug 2020

Dear Dr. Mogi,

Thank you very much for your e-mail and review of the manuscript PONE-D-20-10789 that we sent on April 15, 2020. We thank two reviewers for providing constructive comments regarding the improvement of the original manuscript. We have carefully reviewed the comments and have revised the manuscript accordingly. Our responses are given in a point-by-point manner below. 

We remain very enthusiastic about publishing our original scientific article in the PLOS ONE, and look forward to your editorial decision. 

Sincerely,

Yosuke Yamamoto, MD, PhD

(on behalf of all the authors)

---

## [Decision Letter · Decision Letter 1]

21 Aug 2020

PONE-D-20-10789R1

Longitudinal association between mental health and future antibiotic prescriptions in healthy adults: Results from the LOHAS

PLOS ONE

Dear Dr. Yamamoto,

Thank you for submitting your manuscript to PLOS ONE. After careful consideration, we feel that it has merit but does not fully meet PLOS ONE’s publication criteria as it currently stands. Therefore, we invite you to submit a revised version of the manuscript that addresses the points raised during the review process.

The authors well responded to the Reviewers' comments. However, minor revision is still necessary in the present form. Respond the comments appropriately.

We look forward to receiving your revised manuscript.

Kind regards,

Masaki Mogi

Academic Editor

PLOS ONE

Reviewers' comments:

Reviewer's Responses to Questions

**Comments to the Author**

1. If the authors have adequately addressed your comments raised in a previous round of review and you feel that this manuscript is now acceptable for publication, you may indicate that here to bypass the “Comments to the Author” section, enter your conflict of interest statement in the “Confidential to Editor” section, and submit your "Accept" recommendation.

Reviewer #1: (No Response)

2. Is the manuscript technically sound, and do the data support the conclusions?

Reviewer #1: Yes

3. Has the statistical analysis been performed appropriately and rigorously? 

Reviewer #1: Yes

4. Have the authors made all data underlying the findings in their manuscript fully available?

Reviewer #1: Yes

5. Is the manuscript presented in an intelligible fashion and written in standard English?

Reviewer #1: Yes

6. Review Comments to the Author

Reviewer #1: Thank you for addressing all my comments.

Regarding the selection of the study population (and the included flowchart, sorry for having missed that in my first review), it would be useful to know how many were invited to participate. According to the manuscript, 984 consented to participate, but it is unknown how many declined, and this is important to assess potential selection bias. Please refer to Fig 1 also in the methods section when study population is described.

Thank you for adding text about the internal validity in the discussion. However, I think further explanations are required. I think the sentence “Selection bias might have occurred because not all residents participated” should he complemented a little bit. How many people agreed or not to participate (see my previous point)? If most people participated, then selection bias is unlikely. Do you think participation rates will be similar among those with/without antibiotic use? Or those with/without mental disorders? Selection is likely to cause bias only if associated with both exposure and outcome. Regarding information bias, do you think potential misclassification of the outcome will be differential (different among exposed and unexposed) or non-differential? If non-differential, the bias is likely to be towards the null. I think the manuscript will benefit from such thoughts in the discussion section.

7. PLOS authors have the option to publish the peer review history of their article (what does this mean?). If published, this will include your full peer review and any attached files.

Reviewer #1: **Yes: **Oleguer Plana-Ripoll

---

## [Author Response · Author response to Decision Letter 1]

17 Sep 2020

Thank you for your constructive comments. Unfortunately, we could not know the exact number of residents who refused. Invitations to our study were sent to all residents who received an annual government health check-up, but the exact number was unknown. We added such explanation in our discussion. (P17 L373) We also revised discussion with regard to measurement bias. (P17 L377) We considered the impact of measurement bias on this study was relatively small because exposure (mental health) was not associated with misclassification of antibiotic prescription.

---

## [Decision Letter · Decision Letter 2]

23 Sep 2020

Longitudinal association between mental health and future antibiotic prescriptions in healthy adults: Results from the LOHAS

PONE-D-20-10789R2

Dear Dr. Yamamoto,

We’re pleased to inform you that your manuscript has been judged scientifically suitable for publication and will be formally accepted for publication once it meets all outstanding technical requirements.

Kind regards,

Masaki Mogi

Academic Editor

PLOS ONE

Additional Editor Comments (optional):

No further comment.

Reviewers' comments:

Reviewer's Responses to Questions

**Comments to the Author**

1. If the authors have adequately addressed your comments raised in a previous round of review and you feel that this manuscript is now acceptable for publication, you may indicate that here to bypass the “Comments to the Author” section, enter your conflict of interest statement in the “Confidential to Editor” section, and submit your "Accept" recommendation.

Reviewer #1: All comments have been addressed

2. Is the manuscript technically sound, and do the data support the conclusions?

Reviewer #1: Yes

3. Has the statistical analysis been performed appropriately and rigorously? 

Reviewer #1: Yes

4. Have the authors made all data underlying the findings in their manuscript fully available?

Reviewer #1: Yes

5. Is the manuscript presented in an intelligible fashion and written in standard English?

Reviewer #1: Yes

6. Review Comments to the Author

Reviewer #1: (No Response)

7. PLOS authors have the option to publish the peer review history of their article (what does this mean?). If published, this will include your full peer review and any attached files.

Reviewer #1: **Yes: **Oleguer Plana-Ripoll

---

## [Editor Report · Acceptance letter]

25 Sep 2020

PONE-D-20-10789R2 

Longitudinal association between mental health and future antibiotic prescriptions in healthy adults: Results from the LOHAS 

Dear Dr. Yamamoto:

I'm pleased to inform you that your manuscript has been deemed suitable for publication in PLOS ONE. Congratulations! Your manuscript is now with our production department. 

Kind regards, 

on behalf of

Dr. Masaki Mogi 

Academic Editor

PLOS ONE